# The Antitumor Potential of Sicilian Grape Pomace Extract: A Balance between ROS-Mediated Autophagy and Apoptosis

**DOI:** 10.3390/biom14091111

**Published:** 2024-09-03

**Authors:** Federica Affranchi, Diana Di Liberto, Marianna Lauricella, Antonella D’Anneo, Giuseppe Calvaruso, Giovanni Pratelli, Daniela Carlisi, Anna De Blasio, Luisa Tesoriere, Michela Giuliano, Antonietta Notaro, Sonia Emanuele

**Affiliations:** 1Department of Biological, Chemical and Pharmaceutical Sciences and Technologies (STEBICEF), University of Palermo, Viale delle Scienze, 90128 Palermo, Italy; federica.affranchi@unipa.it (F.A.); antonella.danneo@unipa.it (A.D.); giuseppe.calvaruso@unipa.it (G.C.); anna.deblasio@unipa.it (A.D.B.); luisa.tesoriere@unipa.it (L.T.); michela.giuliano@unipa.it (M.G.); 2Department of Biomedicine, Neurosciences and Advanced Diagnostics (BIND), Institute of Biochemistry, University of Palermo, Via del Vespro 129, 90127 Palermo, Italy; diana.diliberto@unipa.it (D.D.L.); marianna.lauricella@unipa.it (M.L.); giovanni.pratelli@unipa.it (G.P.); daniela.carlisi@unipa.it (D.C.)

**Keywords:** grape pomace, polyphenols, antioxidants, redox status, autophagy/apoptosis balance

## Abstract

From the perspective of circular economy, it is extremely useful to recycle waste products for human health applications. Among the health-beneficial properties of bioactive phyto-compounds, grape pomace represents a precious source of bioactive molecules with potential antitumor properties. Here, we describe the effects of a Sicilian grape pomace hydroalcoholic extract (HE) in colon and breast cancer cells. The characterization of HE composition revealed the predominance of anthoxanthins and phenolic acids. HE treatment was more effective in reducing the viability of colon cancer cells, while breast cancer cells appeared more resistant. Indeed, while colon cancer cells underwent apoptosis, as shown by DNA fragmentation, caspase-3 activation, and PARP1 degradation, breast cancer cells seemed to not undergo apoptosis. To elucidate the underlying mechanisms, reactive oxygen species (ROS) were evaluated. Interestingly, ROS increased in both cell lines but, while in colon cancer, cells’ ROS rapidly increased and progressively diminished over time, in breast cancer, cells’ ROS increase was persistent up to 24 h. This effect was correlated with the induction of pro-survival autophagy, demonstrated by autophagosomes formation, autophagic markers increase, and protection by the antioxidant NAC. The autophagy inhibitor bafilomycin A1 significantly increased the HE effects in breast cancer cells but not in colon cancer cells. Overall, our data provide evidence that HE efficacy in tumor cells depends on a balance between ROS-mediated autophagy and apoptosis. Therefore, inhibiting pro-survival autophagy may be a tool to target those cells that appear more resistant to the effect of HE.

## 1. Introduction

In recent decades, the interest in the use of medicinal plants and phyto-compounds has considerably increased, since it is widely recognized that phytochemicals exert a beneficial effect on human health [1,2]. Scientific evidence has been provided that different natural compounds display biological activities and therapeutic effects that can be compared to those of synthetic drugs [3,4]. A significant amount of the literature reports that bioactive phyto-compounds isolated from natural products are able to counteract the accumulation of reactive oxygen species (ROS), modulating antioxidant enzymes [5]. For this reason, these compounds have a high impact in reducing cellular aging, and they exert anti-inflammatory and antimicrobial actions [6,7]. Moreover, due to their protective properties, natural compounds also potentiate the effects of synthetic molecules in the prevention of cancer [8,9]. In light of these benefits, the term “functional foods” has been coined, referring to foods that contain molecules with positive effects on health beyond basic nutritional values [10]. The health beneficial properties mentioned above are mainly attributable to some specific components of functional foods including alkaloids, terpenoids, and polyphenols.

Phenolic compounds, which include flavonoids and non-flavonoids, represent a class of molecules with a common chemical structure characterized by hydroxyl groups associated with benzene rings. These compounds can be used for the prevention and treatment of many pathologies, including cancer [11,12]. However, apart from their antioxidant effects, the studies on their biological actions are fragmentary and not fully identified from a biochemical point of view [13]. 

In recent years, the need to identify and recover more bioactive compounds has also led to the recovery and valorization of large amounts of waste from the main agri-food industries [14]. The principle of recovery and valorization of waste products perfectly fits into the circular economy perspective, a high performing production model that tends to extend the life cycle of products, helping to reduce waste and giving them a new application value [15].

In this context, grape pomace, which consists of grape seeds, skins, and stalks generated by the wine-making industry, represents a precious source of bioactive molecules. Increasing interest in the reuse and valorization of grape pomace is not only focused on keeping the amount of waste production worldwide under control (8.49 million tons) [16], but especially on providing health benefits due to its anticancer, antioxidant, anti-inflammatory, anti-aging, and antimicrobial potential [17].

Many studies have highlighted the different beneficial properties of bioactive compounds present in grape pomace [18,19]. Interestingly, grape pomace derived-polyphenols seem to be also promising in cancer prevention and treatment. Research studies have presented findings obtained in various cancer models. For instance, resveratrol, a well-known polyphenol present in grape pomace, was shown to be able to arrest cell proliferation and induce cell death in DU145 prostate cancer cells [20]. Phenolic compounds of Aglianico grape pomace demonstrated anti-proliferative effects on HT29 and SW480 cell lines, which were accompanied with a regulation of Bcl-2 family members and activation of caspase-3 [21]. Moreover, recent data indicate the ability of grape pomace to positively modulate both normal and dysbiotic gut microbiota [22], providing a rationale for its application in pharmaceutical or cosmetic industries [23,24]. 

It is important to consider that the composition of the biomolecules in grace pomace is strongly dependent on several factors, including: the specific cultivar, the cultivation methods, the pedoclimatic conditions, and the harvest period. 

Considering these premises, in this study, we evaluated for the first time the potential of a variety of Pinot Gris grape grown in the western area of Sicily. This region has an ancient tradition in the wine industry; archaeological discoveries suggest that wine production in Sicily is among the oldest attested in the world, dating back to at least 6000 years ago [25]. Today, more than one hundred thousand hectares are allocated to produce millions of hectoliters of wine in Sicily.

This paper specifically describes the antitumor effects of grape pomace extract on HCT116 colon cancer and MDA MB-231 breast cancer cell lines. These models were chosen as representatives of two of the most common tumors worldwide. In Italy, colorectal cancer rank as the third most commonly diagnosed cancer in men (14%), and as the second most commonly diagnosed cancer in women (12%) [26]. Meanwhile, breast cancer accounts for the highest diagnosis rate among women (30%). Nowadays, after surgical removal, when possible, the elective treatments of both cancer types are represented by chemotherapy, which is often described as cytotoxic for normal cells and producing several side effects in the organism. Therefore, the identification of natural phyto-compounds derived from agri-food wastes, potentially inducing cell death, can provide an adjuvant for conventional therapy in order to reduce side effects [27].

The results reported in this study demonstrate the antitumoral potential of Pinot Gris pomace extracts in colon and breast cancer cells. Specifically, the paper describes the involvement of oxidative stress and autophagic cell death induced by a Sicilian grape pomace hydro-alcoholic extract. 

## 2. Materials and Methods

### 2.1. Grape Pomace Extract Preparation

The crude extract was prepared from white grape pomace of Vitis vinifera cultivar “Pinot gris” from the 2023 harvest, provided by Testa-Ferrarella’s agri-food industry, (Trapani, Sicily, Italy). Sun-dried grape pomace (grape skin, seeds, and stem) was maintained at −80 °C until use. Before extraction, the matrix was ground to powder. Preliminary tests were carried out to select the most appropriate chemical–physical conditions to optimize the extraction method. An organic solvent (n-hexane) was used in a ratio of 1:5 (*w*/*v*) to eliminate the lipid components. The dried delipidated matrix was extracted using 57:43 (*v*/*v*) ethanol:water as a solvent at a solid-to-liquid ratio of 1:5 (*w*/*v*) for 60 min at 58 °C to isolate the polyphenolic compounds from grape pomace. Then, the extract was subjected to centrifugation at 5000 rpm for 15 min and then at 13,000 rpm for 20 min at 4 °C. The grape pomace hydroalcoholic extract (HE) was filtered with a 0.22 µm filter and stored at −20 °C until use. The dry weight of HE was determined after lyophilization and was 62.5 mg/g of grape pomace. 

### 2.2. Assessment of the Total Polyphenolic Content 

The concentration of total polyphenols of HE was determined by using the Folin–Ciocalteu (Merk-Sigma Aldrich, Milan, Italy) colorimetric assay, employing a curve of gallic acid as external calibration [28]. Briefly, 100 µL of Folin–Ciocalteu reagent was added to 100 µL of the extract (diluted 1:100 in distilled water). After adding 800 µL of 5% sodium carbonate, the samples were left at 40 °C for 20 min. Then, the samples were rapidly cooled on ice and the absorbance was measured spectrophotometrically at 765 nm. The concentration of the total polyphenols was estimated as µg of gallic acid equivalent (GAE) by using an equation obtained from gallic acid calibration curve. In the experiments, we used concentrations of HE referred to polyphenols content expressed as µg GAE/mL.

### 2.3. Determination of Phenolic Compounds by UHPLC-Orbitrap-MS

UHPLC-Q Exactive Orbitrap-HRMS system (Thermo Fisher Scientific™, Bremen, Germany) composed of a Dionex Ultimate 3000 liquid chromatograph coupled to a Q Exactive™ Plus Hybrid Quadrupole-Orbitrap™ Mass Spectrometer equipped with a heated electrospray ionization (HESI) ion source (Headquarters Waltham, MA, USA), was used to analyze grape pomace extract sample.

Chromatographic separation was achieved on Luna C18(2) (150 × 2.0 mm, 5 µm) equipped with precolumn, with 0.1% formic acid in water (mobile phase A) and 0.1% formic acid in methanol (mobile phase B). A gradient method at 200 μL/min flow rate was applied as follows: start at 5% B, stay for 2 min; increase to 95% B over 18 min, held for 2 min; then decrease to 5% B over 18 min; and maintained constant for 2 min for a total run time of 40 min. Injection volume was 1 μL.

A full mass and targeted SIM (t-SIM) scan methods were applied. The Orbitrap parameters were set as follows: negative (−) ESI full scan mode and t-SIM, sheath gas flow rate 30 AU, discharge voltage 3.0 kV, capillary temperature 300 °C, resolution 35,000 FWHM, AGC target 5 × 10^6^, maximum injection time 200 ms, and scan range 80–1000 *m*/*z*.

Calibration curves were constructed at five calibration levels for: *p*-hydroxybenzoic acid, protocatechuic acid, *p*-coumaric acid, gallic acid, kaempferol, and quercetin. When reference compounds were not available, the calibration of structurally related substances was used. 

### 2.4. Cell Cultures and Treatment Conditions

Colon (HCT116) and breast cancer (MDA MB-231) cells (provided by Interlab Cell Line Collection, (ICLC, National Institute of Cancer Research, Genoa, Italy) were used to test the biological effects of HE. Cells were cultured in Dulbecco’s modified Eagle’s Medium (DMEM) enriched with 2 mM glutamine and 10% heat-inactivated fetal bovine serum (FBS). A solution of 100 U/mL penicillin and 100 µg/mL streptomycin was added to the medium and 1% non-essential amino acids only for breast cancer cells. Cells were incubated at 37 °C in the presence of 5% CO_2_ and in a humidified atmosphere. 

For the experiments, cells were seeded in opportune culture plates at 70–80% confluence for western blotting and MTT assay or 40–50% for cell cycle and fluorescence analysis. After overnight incubation, cells were treated for the selected times. *N*-acetyl cysteine (NAC) and bafylomicin A1 (BafA1) were used at the concentration of 5 mM and 100 nM, respectively, pre-treating cells for 1 h before adding HE. Cells unexposed to the treatment were incubated with the vehicle alone used as controls and analyzed at the same time intervals.

### 2.5. Cell Viability Assays

Cell viability was evaluated after treatment with different concentrations of HE for the indicated time. The colorimetric assay using MTT (3-(4,5-dimethylthiazol-2-yl) -2,5-diphenyltetrazolium bromide) (Merk, Milan, Italy) was used to measure cellular metabolic activity as an indicator of cell viability and proliferation [29]. Briefly, cells were plated in 96-well plates in the presence of 200 µL of culture medium and, after overnight incubation, the compounds were added. At the end of treatment, 20 µL of a MTT solution (5.5 µg/mL) was added. The formazan crystals were dissolved in lysis solution (20% sodium dodecyl sulfate, 40% *N*,*N*-dimethylformamide, pH 4.7) and the staining was quantified by measuring the absorbance at 570 nm at a microplate reader (OPSYS MR, Dynex Technologies, Chantilly, VA, USA). The measured absorbance is directly proportional to the number of viable cells. The viability of treated cells was expressed as the percentage of the optical density (OD) values compared to control cells.

Cell viability was also analyzed by propidium iodide (PI) staining. For analysis, cells were seeded in 24-well plates in the presence of 1 mL of culture medium. After adhesion and treatments, cells were washed in PBS (Phosphate Buffered Saline) and incubated with PI (1 µg/mL) for about 10 min in the dark at room temperature as reported [30]. Images were acquired at magnification of 200× by using an Optika microscope (OPTIKA S.r.l., Ponteranica (BG), Italy) endowed with a computer imaging system (OPTIKA PROVIEW, version x64, 4.11.20805.20220506).

### 2.6. Cell Cycle Analysis

To assess the distribution of cell cycle phases, cells were collected after HE treatment by trypsinization (0.025% trypsin-EDTA; Life Technologies Ltd., Monza, Italy) and then were resuspended in a hypotonic solution (25 µg/mL propidium iodide, 0.1% sodium citrate, 0.2% Igepal, and 10 µg/mL RNase A). The distribution of cells on the different cycle phases was analyzed using a Cytoflex Flow Cytometer (Beckman Coulter Life Sciences Italia, Milan, Italy) equipped with CytoExpert flex software (version 2.5.0.77, Beckman Coulter). Cell debris and aggregates were eliminated through appropriate gating. The index of DNA fragmentation was estimated by evaluating the percentage of cells in subG0/G1 phase. At least 3 × 10^4^ cells per sample were analyzed and data were saved in list mode files. The results presented in the figures are representative of three independent experiments. 

### 2.7. Western Blotting Assay and Antibodies

Western blotting analyses were performed to evaluate the expression of proteins involved in activated pathways. Briefly, cells were detached and lysed with RIPA cell lysis buffer (1% Igepal, 0.5% sodium deoxycholate, 0.1% SDS, and protease inhibitors, pH 7.4) at 4 °C to preserve cellular proteins from degradation. A known amount of protein extract (30 µg), evaluated by means of a Bradford Protein Assay (Bio-Rad Laboratories S.r.l., Milan, Italy), was subjected to separation on polyacrylamide gel under denaturing conditions. After electrophoresis, proteins were transferred to a nitrocellulose filter through electroblotting. Immunodetection was then performed by incubating filters with specific primary antibodies against the proteins of interest. In detail, anti-HO-1 (GTX101147) was purchased from Gene Tex (GeneTex, Prodotti Gianni, Milan, Italy); anti SOD-2 (sc-133), p53 (sc-126), Nrf2 (Sc-722), AKT (sc-8312), p21 (sc-6246) and γH2AX (Sc-517348) from Santa Cruz Biotechnologies (Santa Cruz, CA, USA); anti p62 (P0068) and actin (A4700) from Merk Millipore (Merk-Sigma Aldrich, Milan, Italy); anti Caspase-3 (96625), PARP1 (9542); LC3 (2775S) from Cell Signaling (Cell Signaling Technology, Danvers, MA, USA), and phospho-AKT, S473, (AF887) from an RD system. Antibodies signal was developed through HPR-conjugated secondary antibodies (Promega, Fisher Scientific, Rodano, Milan, Italy) and revealed by using enhanced chemiluminescence (ECL) reagents (Amersham, Fisher Scientific, Milan, Italy) and ChemiDoc XRS (Bio-Rad Laboratories, Hercules, CA, USA). In some experiments, Western blotting membranes were washed and stripped to remove the primary and the secondary antibody complex and re-probed with a second pair of antibodies to allow the detection of another target protein. Original figures can be found in Appendix A. 

### 2.8. Reactive Oxygen Species Detection

The reactive oxygen species (ROS) were detected through the oxidation of the cell-permeant 2′,7′-dichlorodihydrofluorescein diacetate (H_2_DCFDA) (Molecular Probe, Life Technologies, Eugene, OR, USA) dye, as reported before [31]. Cells were seeded in 24-well plates at the density of (2 × 10^4^/cm^2^); after 3, 6, and 24 h of treatment with two different concentrations of HE alone or in combination with NAC. Cells were washed in PBS and incubated with 2 μM of H_2_DCFDA dye for 15 min in the dark and in the presence of 5% CO_2_ at 37 °C. At the end of incubation, the excess fluorochrome was removed, the wells were washed in PBS, and the fluorescent 2’,7’-dichlorofluorescein (DCF) produced by intracellular oxidation was analyzed with fluorescence microscope using FITC filter (Optika Science, Srl, Ponteranica, Italy). Images were acquired at magnification of 200× by using an Optika IM3FL4 fluorescence microscope equipped with a computer imaging system (OPTIKA PROVIEW, version x64, 4.11.20805.20220506).

### 2.9. Evaluation of Autophagic Vacuoles Formation

The formation of acidic vesicular organelles was observed using monodansylcadaverine (MDC) staining. Cells were treated with 0.05 mM MDC in PBS. After an incubation at 37 °C for 15 min, cells were washed three times with PBS and analyzed under fluorescence microscope using DAPI filter (Optika Science, Srl, Ponteranica, Italy). Images were acquired at magnification of 200× by using an Optika IM3FL4 fluorescence microscope equipped with a computer imaging system (OPTIKA PROVIEW, version x64, 4.11.20805.20220506).

### 2.10. Statistical Analysis

Data were presented as mean ± standard deviation (SD). Analyses were carried out using the Student’s *t*-test evaluated with Excel software, (Microsoft Office 365). Comparisons were made between the control (untreated) vs. all treated samples by employing a one-way ANOVA test, followed by post-hoc Bonferroni’s test. *p* values less than 0.05 were considered significant.

## 3. Results

### 3.1. Chemical Characterization of the Hydroalcoholic Grape Pomace Extract (HE)

In the present study, a hydroalcoholic extract (HE) from Sicilian grape pomace was prepared and tested to determine whether it displays possible antitumor effects in colon and breast cancer cells. Initially, the content of total phenolic compounds was analyzed in the extract prepared from the last harvest by using a colorimetric assay. The results showed that HE contained 30 mg ± 5 GAE/g of dried grape pomace in accordance with the literature data concerning total phenolic content from white grape pomace [32]. However, although total phenolic values were in broad agreement with those reported by other authors, it is well known that they are closely related to the grape cultivar, geographical origin, winemaking practices, and extraction methodology. Therefore, we analyzed and quantified the main components of HE using high-resolution mass spectrometry.

The analysis identified the presence of 17 phytochemicals out of 23 tested with their relative retention times (RTT), exact masses, and concentrations expressed in mg of the analyte per 100 g dry weight (DW) (Table 1). We found that the main components of Sicilian grape pomace HE include: catechin, epicatechin, and quercetin within the anthoxanthins group (91.96, 52.74, 20.04 mg/100 g DW), and gallic acid among the phenolic acids (20.96 mg/100 g DW). 

### 3.2. The Effects of the HE Extracts on HCT116 and MDA MB-231 Cell Viability and Cell Cycle Distribution 

Grape pomace HE was tested to determine its ability to modulate the viability of colon (HCT116) and breast (MDA MB-231) cancer cells. First, possible cytotoxic effects of HE were assessed by MTT assay using increasing concentrations of the extract. As shown in Figure 1a, HE reduced cancer cell viability in a dose-dependent manner when employed at concentrations higher than 25 µg GAE/mL. The reducing effect was more evident in HCT116 cells, where a reduction of about 70 and 90% was observed with 75 µg GAE/mL HE for 48 and 72 h, respectively. Cytotoxic effects appeared to be less evident in breast cancer cells, where the reduction of vitality did not exceed 40% even with a long treatment time (72 h). 

Propidium iodide (PI) staining confirmed these results. As reported in Figure 1b, HCT116 cells were dramatically reduced in number and showed significant PI positivity after HE treatment at 75 µg/mL for 72 h. On the other hand, with the same treatment conditions, MDA MB-231 cells were less sensitive to the effects of the extract. Indeed, viable cells were much more represented after treatment and only a few cells were PI positive. 

Based on these results, the higher concentrations of HE (75 and 150 µg GAE/mL) were employed to evaluate the effect on cell cycle distribution of both cell lines after 48 h treatment (Figure 2). HE treatment increased the percentage of HCT116 cells in subG0-G1 phase of the cell cycle in dose- and time- dependent manner reaching a value of 33.98% vs. 3.32% in control cells, indicating DNA fragmentation at 48 h. Conversely, HE did not produce a subG0-G1 peak in MDA MB-231 cells but only caused an increase in the G1 phase percentage (from 70.33% in untreated cells to 80.22% in HE-treated), indicating a cytostatic effect.

Considering the diversified effects of HE on HCT116 and MDA MB-231 cells, to investigate the molecular pathways induced by HE in the two cell lines, we focused on proteins involved in apoptosis, cell cycle arrest, and cell survival. First, we evaluated the expression level of two apoptotic markers, caspase-3, a critical protease executioner of apoptosis, and PARP1, the main substrate of activated caspase-3. Western blot analysis was carried out after 48 h treatment, a proper time in which the reducing effects on cell viability were quite evident but not dramatic. The results showed a considerable reduction of pro-caspase-3 in HCT116 cells, which was interpreted as the enzyme activation (Figure 3, left panel). The activation of caspase-3 was confirmed by the cleavage of PARP1, a well-known caspase 3 substrate [33], with the appearance of the canonical 89 kDa fragment (Figure 3, left panel). Differently, the caspase-3/PARP1 axis seemed not to be involved in MDA MB-231 cells (Figure 3, right panel), where the reduction in cell viability was mainly due to a cytostatic effect, as previously evidenced. In particular, significant variations in the level of pro-caspase-3 were not observed in MDA MB-231 cells, while, concerning PARP1, it was only detected by the appearance of the 66 kDa fragment, a cleavage product that is reported to result by matrix metalloproteinase-2 action [34]. 

It is well known that apoptosis induction is often associated with a modulation of pro-survival factors. In this regard, to detect possible differences in the two cell lines, we analyzed γH2AX (phosphorylated form of histone H2AX at Ser-139) as a marker of DNA double strand breaks which is involved in the first phase of the apoptotic pathway [35]. After HE treatment, γH2AX level was markedly increased in HCT116 cells, while it was unaffected in MDA MB-231 cells (Figure 4). 

We also considered p21/WAF1, which is usually associated with cell cycle arrest, but it has been recently characterized by a controversial role in cancer cells, including a pro or anti-apoptotic one [36]. p21 level resulted remarkably increased in colon cancer cells while only a modest increase was found in MDA MB-231 cells. Lastly, the level of AKT, one of the most important cellular pro-survival factors, was also evaluated. In HCT116 cells, a remarkable decrease in both the levels of AKT and its phosphorylated (active) form (pAKT) was observed. In contrast, in MDA MB-231 cells a significant increase in the pAKT level was found (Figure 4).

Taken together, these results suggest that HE was able to induce a canonical apoptotic pathway in HCT116 but not in MDA MB-231 cells.

### 3.3. The Effects of HE Were Associated with Reactive Oxygen Species (ROS) Production

It is well known that tumor cells produce moderate/high levels of ROS which can sustain tumor promotion and progression. On the other hand, ROS can induce tumor cell death triggering different death programs, including apoptosis [7,37]. 

To investigate the molecular mechanisms underlying HE-induced effects on HCT116 and MDA MB-231 cells and a possible role of ROS, we monitored intracellular ROS production along time with different doses of HE alone or in combination with *N*-acetyl-cysteine (NAC), a widely known antioxidant [38].

Figure 5 shows ROS production after 6 h of HE treatment, visible as the increase in the green fluorescence due to the conversion of nonfluorescent H_2_DCFDA to fluorescent dye DCFDA. It is possible to observe ROS production in both HCT116 and MDA MB-231 cells (Figure 5a), however, it was more evident in HCT116 cells. The effect was prevented by NAC in both cell lines at this treatment time. Moreover, time-course experiments evidenced that in HCT116 cells, the ROS increase was a very precocious event, while it markedly decreased at values comparable to the control after 24 h treatment. Also in this case, a different behavior of MDA MB-231 was observed. Indeed, in these cells, ROS relative fluorescence was maintained high until 24 h (Figure 5b). Moreover, the presence of NAC almost completely counteracted HE-induced ROS increase in HCT116 cells while it exerted a minor effect in MDA MB-231, especially at 24 h, the time where the maximum ROS level was detected in this cell line (Figure 5b).

The decrease of ROS in HCT116 cells at a long treatment time (24 h) was most likely the result of a cellular response to the insult due to antioxidant defenses. As a confirmation, we analyzed the levels of some well-known antioxidant factors. As shown in Figure 6, after 24 h treatment, in HCT116 cells a remarkable increase in the expression level of heme oxygenase-1 (HO-1), the primary enzyme which counteracts oxidative injury [39], was observed. HO-1 increase was probably correlated with the activation of Nrf2, the well documented transcription factor promoting its expression [40]. As expected, Nrf2 levels concomitantly increased following HE treatment. At the same time, SOD-2, a protein that binds the superoxide by product [41], strongly raised its expression (six-fold respect untreated cells) after 24 h of HE exposure. These effects were completely (in the case of HO-1 and Nrf2) or partially (SOD-2) counteracted when HE treatment was preceded by incubation with the antioxidant NAC (Figure 6). 

In MDA MB-231 cells, only a moderate increase in the expression of the antioxidant factors and a reduced preventive effect of NAC were observed. These data are in accordance with the persistence of high ROS levels in these cells at 24 h. 

### 3.4. ROS Increase Is Linked to the Induction of an Autophagic Pro-Survival Pathway

Autophagy is a self-clearance pathway adopted by the cells to remove damaged organelles and proteins, including the oxidized forms. In many cellular systems the increase in ROS levels is related with the activation of an autophagic mechanism. ROS can induce autophagy, and autophagy, in turn, can promote oxidative stress [42,43,44]. Moreover, we have previously shown a particular sensitivity of colon cancer cells to the induction of an autophagic process [31,45,46].

To analyze the possible induction of an autophagic pathway in our models, we labeled acidic vacuoles by using the fluorescent dye monodansylcadaverine (MDC). Images reported in Figure 7a show that HE increased the number of acidic vacuoles in both HCT116 and MDA MB-231 cells. This effect was already evident after 6 h of treatment, being HCT116 more responsive than MDA MB-231 cells. The effect progressively diminished in a time-dependent manner, with fluorescence values similar to the controls after 48 h of treatment (Figure 7b).

The induction of autophagy was confirmed by the evaluation of the specific autophagic markers p62 and LC3 by western blot analysis. LC3 (microtubule-associated protein 1 light chain 3) is a protein recruited from the cytosol (LC3-I) to the membrane of autophagosomes where it is cleaved and lipidated (LC3-II), while p62/SQSTM1 behaves as an adaptor protein that serves as a link between LC3 and ubiquitinated substrates [47].

We performed time and dose-dependent experiments to monitor the trend of these markers after HE treatment. Both p62 and LC3 levels increased at 24 h of HE treatment in the two cell lines. However, while at 48 h the levels of the proteins decreased to values near to the control in HCT116 treated cells, the levels of the two autophagic markers remained high in MDA MB-231 cells (Figure 8).

Pretreatment with bafilomycin A1 (BafA1), a potent inhibitor of the autophagic flux [48], clearly reduced HE effects on autophagic vacuoles formation (Figure 7), maintaining high the levels of autophagic markers at the same time (Figure 9). 

Interestingly, bafilomycin A1 pretreatment did not modify cell viability of HE-treated HCT116 cells, while it significantly increased the effect of HE in MDA MB-231 indicating that autophagy is activated as a pro-survival response in these cells (Figure 10a). Moreover, the activation of apoptotic cell death pathway was not observed in MDA MB-231 cells, as also indicated by no variations in the pro-caspase-3 level in bafilomycin A1 HE-pre-treated cells (Figure 10b). 

Finally, the effect of HE on the levels of p62 and LC3-II was prevented by the antioxidant NAC, confirming a relationship between ROS production and the triggering of the autophagic flux (Figure 9). 

## 4. Discussion

The various biological activities of grape pomace (antioxidant, antimicrobial, prebiotic, anti-proliferative), have been widely described in the literature sustaining a high impact on human health and a plethora of possible applications [2,3]. Moreover, specific studies on the health potential of waste products represent a new frontier of applied research. This is a goal of circular economy to avoid food waste accumulation, which is recognized as responsible for serious environmental and economic concerns [49]. 

This paper specifically describes the antitumor effects of a hydroalcoholic extract (HE) from white grape pomace of Sicilian Vitis vinifera “Pinot gris” cultivar from the 2023 harvest. In our conditions and using the extraction parameters that we have chosen, the extract resulted particularly rich in anthoxanthins, such as catechin, epicatechin and quercetin. Gallic acid was also quantified as the most representative phenolic acid. 

The study was mainly focused on the biochemical mechanisms that account for the anti-proliferative and pro-apoptotic action of HE from Sicilian grape pomace. We found a different response in two cell lines, HCT116 colon cancer and MDA MB-231 breast cancer cells, that were originally chosen as representative models of two tumor types among the most widespread. Although these cell lines profoundly differ being of different origin, they allowed to detail the biochemical execution of apoptosis and autophagy, two main processes involved in the antitumor response [50], following HE treatment. 

In view of circular economy, we have previously reported that HCT116 cells represent a good model to study the effects of natural compounds and respond to a crude Mango fruit peel extract, cultivated in Sicilian territory [35,51]. 

Concerning colon cancer cells, we showed that HCT116 cells are very sensitive to HE antitumor action and undergo apoptosis by shifting the balance between pro-survival and pro-apoptotic factors toward the second ones. To our knowledge, this represents the first study on this highly aggressive colon cancer model concerning the effects of grape pomace hydroalcoholic extract. The only evidence in the literature was provided by Perez et al. who studied the effects of crude or partially purified extracts of grape pomace from Spanish territory, demonstrating a modest activity in other colon cancer models, CaCo-2 and HT-29 cells [52], reported to display a lower degree of aggressiveness than HCT116 [53]. 

Here we show that HCT116 underwent apoptosis following HE treatment as shown by caspase-3 activation and PARP1 degradation. At the same time, a decrease of pro-survival factor phospho-AKT and a significant increase in the level of γH2AX, a marker of DNA damage, were observed. 

Concerning the breast cancer model, MDA MB-231 cells represent a highly aggressive triple negative breast cancer cell line. Our data show that HE was capable of reducing cell viability also in this case, but neither the morphological features nor the cell cycle distribution seemed to indicate the activation of apoptotic cell death. As a confirmation, no caspase-3 activation was observed, and another pattern of PARP degradation (66 kDa PARP1 fragment) occurred without significant modifications in the levels of γH2AX. Moreover, an increase in the level of the pro-survival phosphorylated AKT was detected. It is well known that AKT is activated by phosphorylation. Usually, two key residues account for AKT phosphorylation and consequent activation: Thr308 in the kinase domain, which is typically phosphorylated first, and Ser473 in the C-terminal tail, which is required for full activation of AKT. Since we used a phospho-AKT antibody that specifically recognizes Ser473, we assumed that phosphorylation in this residue is associated with AKT activation. As mentioned before, it is interesting to note that HE extract produced opposite effects on AKT phosphorylation in the two cell lines, thus suggesting that a pro-survival response occurs in breast cancer cells that appear to be more resistant to the effects of the extract. 

To better characterize the molecular events responsible for the diversified effects on the two tumor models, we focused on oxidative stress and autophagy, two main conditions profoundly related with tumor growth and response to antitumor agents [42,54]. For many years, a debate has arisen on the role that ROS and oxidative stress exert on tumorigenesis and therapy response until reaching the conclusion that ROS are Janus molecules. Indeed, they can behave differently depending on the tumor type, tumor developmental stage, specific pro-oxidant stimuli, and their cellular concentration [55]. It has been shown that ROS content is higher in many tumor cell types compared to the normal counterpart. The dual role of ROS implies that, on one hand, ROS production is necessary to maintain cell survival and to sustain cell proliferation; on the other hand, ROS accumulation overcoming a certain threshold becomes cytotoxic to cancer cells. 

Our data show that HE increased ROS levels in both HCT116 and MDA MB-231 cell lines in the first period of treatment (up to 6 h), thus suggesting that initial oxidative stress occurs. However, while in HCT116 colon cancer cells, ROS level tended to decrease to levels similar to the control within 24 h, in MDA MB-231 breast cancer cells, the levels remained high even at 24 h. The analysis of antioxidant systems, including Nrf2, SOD-2 and HO-1, revealed that HCT116 triggered an early antioxidant response while a lower expression of antioxidant factors was detected in MDA MB-231 cells at the same HE treatment conditions. 

It is possible to speculate that the ROS threshold is different in the two cell lines. The level of ROS could confer a state of increased basal oxidative stress that increases the vulnerability to chemotherapeutic agents that, in turn, augments ROS generation or weakens antioxidant defenses of the cell [56]. While colon cancer cells rapidly underwent oxidative stress with consequent apoptosis, most likely breast cancer cells maintained a high ROS levels to sustain cell survival. Indeed, in MDA MB-231 cells the persistent ROS level, due to a lower expression of antioxidant factors, was not able to induce apoptosis, but only resulted in a cytostatic effect. These findings are in line with the observation of Martino et Al. who found a high basal level of ROS in MDA MB-231 compared to other breast nonmalignant cells [57]. This interpretation was also sustained by our data indicating that the preincubation with the antioxidant NAC, completely prevented HE pro-oxidant action in HCT116 cells, but resulted less efficacious in MDA MB-231 cells. Considering these findings, to verify the pro-survival hypothesis, autophagy was investigated in both cell lines after treatment with the extract. It is well known that autophagy is a double-faced process in tumor cells [58,59]. As a catabolic process, autophagy is usually activated in tumor cells to sustain cell survival. On the other hand, depending on the interplay with apoptosis, autophagy can be a death promoting event and an “Achilles’ heel” in tumor cells. Moreover, a complex connection between ROS and autophagy has been well documented [60,61]. Autophagy can promote ROS accumulation and, vice versa, increased ROS levels can regulate the autophagic process. Here, we report the involvement of autophagy in both cell lines upon HE treatment, as demonstrated by the increase of acidic vacuoles, accompanied by high levels of p62 and LC3II. However, the process evolved in a different way in the two cell lines. In HCT116 cells the level of autophagic markers increased up to 24 h and then decreased at 48 h. These data led us to conclude that autophagy was completed. Therefore, since we showed that in these cells autophagy has a pro-survival role, its completion most likely permits the activation of apoptosis. Moreover, the reduction of autophagy observed in these cells in the second phase of treatment can be correlated with the strong activation of p21. Indeed, a recent study demonstrates that in HCT116 cells the inhibition of p21 resulted in a substantial induction of autophagy [62]. 

Differently in MDA MB-231 cells, the levels of the autophagic markers were maintained very high at 48 h, indicating a still ongoing process most likely sustaining cell survival. This hypothesis was supported by the observation that the addition of bafilomycin A1, an autophagy inhibitor, was able to strongly potentiate the HE effect on MDA MB-231 cell viability, while displaying no significant effect in HCT116 cells. Specifically, pre-treating the cells for one hour with bafilomycin A1 was considered the condition that guaranteed autophagy inhibition as previously described [30]. Moreover, in these conditions, activation of caspase-3 was not observed in MDA MB-231 cells, indicating that HE induced a different death program when autophagy was blocked. Ongoing studies aim to clarify the death mechanism induced by HE in breast cancer cells under autophagy inhibition. 

Overall, we provided evidence that the HE extract displays an antitumor potential which is correlated with a pro-oxidant action. It clearly depends on the tumor type and whether the HE is able to induce classic apoptosis, like in the case of HCT116 colon cancer cells. In other cases, such as that of MDA MB-231 breast cancer cells, the inhibition of pro-survival autophagy may be a good strategy to target those tumors that do not undergo apoptosis but may activate alternative death pathways. 

## 5. Conclusions

Overall, the results presented in this paper align well with the need for recycling of waste products in perspective of circular economy. In particular, our findings provide insight into the ability of grape pomace HE to reduce the proliferation of colon and breast cancer cells. The biochemical analysis of HE signaling demonstrated the activation of a canonical apoptotic pathway in colon cancer cells, while in breast cancer cells, HE exacerbated a survival autophagy which was counteracted by the addition of its specific inhibitor BafilomycinA1. Upstream of these effects, we demonstrated the induction of oxidative stress which was responsible for the activation of the antioxidant response. In summary, HE has the capacity to disrupt the connection between redox equilibrium and autophagy in cancer cells, thereby influencing the fate of these cells.

## Figures and Tables

**Figure 1 biomolecules-14-01111-f001:**
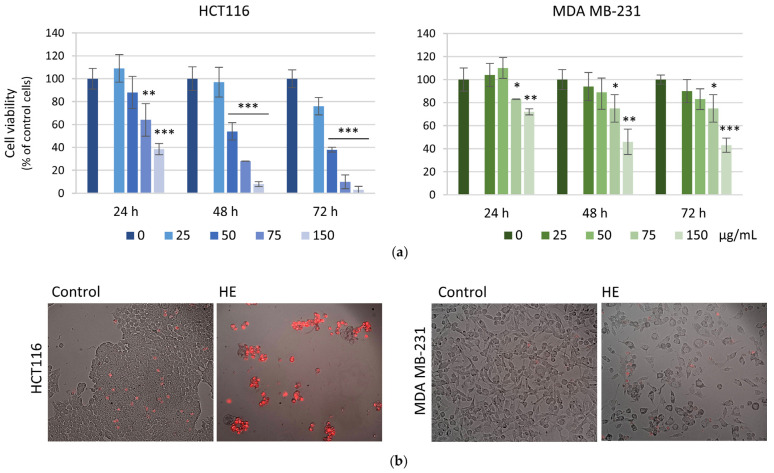
Time and dose-dependent effects of HE on HCT116 and MDA MB-231 cells. (**a**) Cell viability was assessed by MTT assay after 24, 48 and 72 h of HE treatment. HE was used at concentration of 25–150 µg GAE/mL. Results are representative of three independent experiments and expressed as mean ± SD. * *p* < 0.05, ** *p* < 0.01, *** *p* < 0.001 vs. untreated control. (**b**) Representative merged images of bright field and propidium iodide-stained cells after 72 h treatment with 75 µg GAE/mL HE. Cells were visualized under fluorescent microscope at 200× magnification and the pictures acquired by OptiKa Proview software.

**Figure 2 biomolecules-14-01111-f002:**
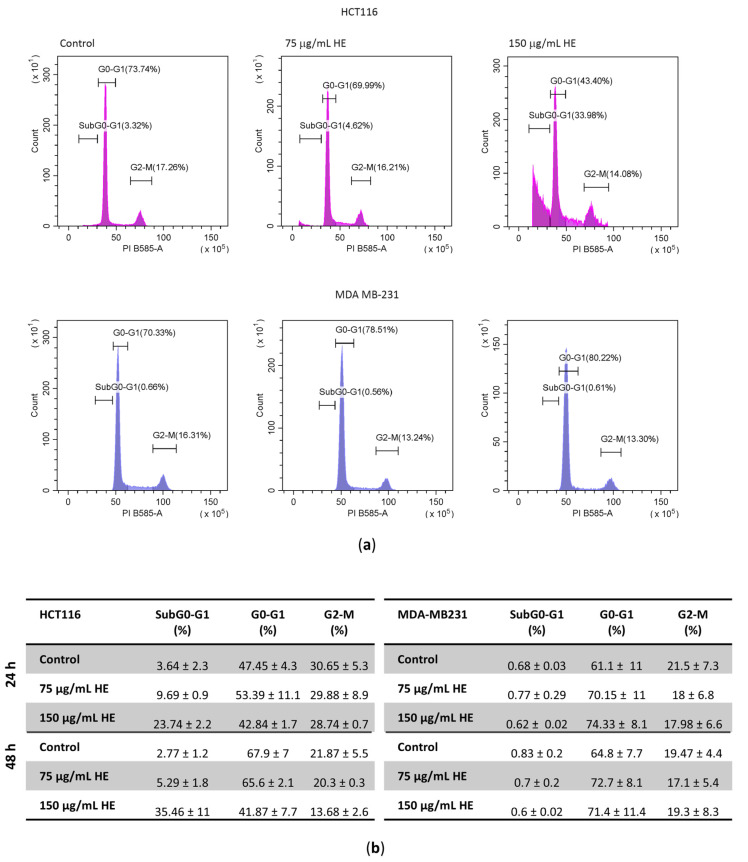
Effects of HE on HCT116 and MDA MB-231 cell cycle distribution. (**a**) Flow cytometric analysis of cell cycle distribution after 48 h of 75 and 150 µg GAE /mL HE treatment. (**b**) Summary table of the percentage of cells in the different phases of cell cycle after 24 and 48 h of treatment with 75 and 150 µg/mL GAE HE. Data are expressed as the mean ± SD.

**Figure 3 biomolecules-14-01111-f003:**
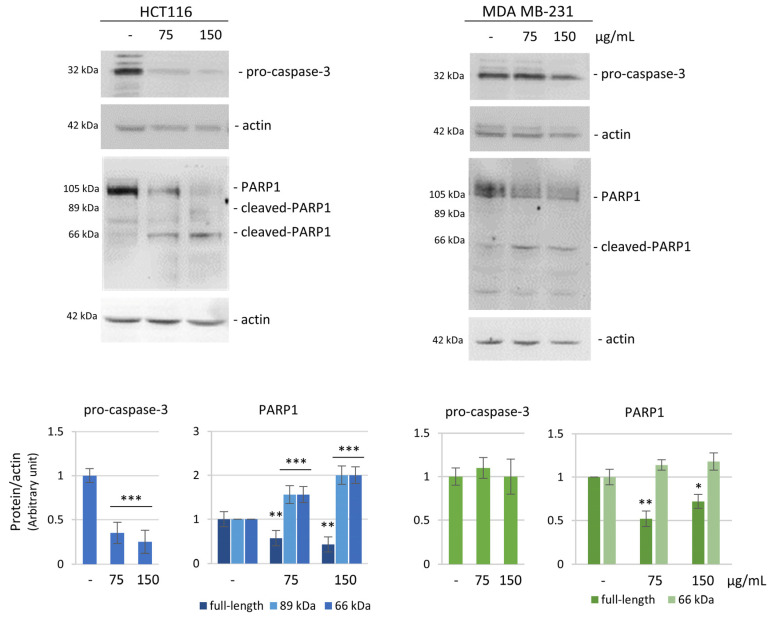
HE effect on the expression of apoptotic markers. Western blotting analysis of pro-caspase-3 and PARP1 after 48 h of 75 and 150 µg GAE/mL HE treatment in HCT116 cells (**left** panel) and MDA MB-231 cells (**right** panel). Actin blot was used as a loading control to normalize the protein levels. Densitometric analysis, conducted by Quantity One software (version 4.6.6 basic), are reported in the histograms below. Results are representative of three independent experiments and values are expressed as mean ± SD. * *p* < 0.05, ** *p* < 0.01, *** *p* < 0.001 vs. untreated control.

**Figure 4 biomolecules-14-01111-f004:**
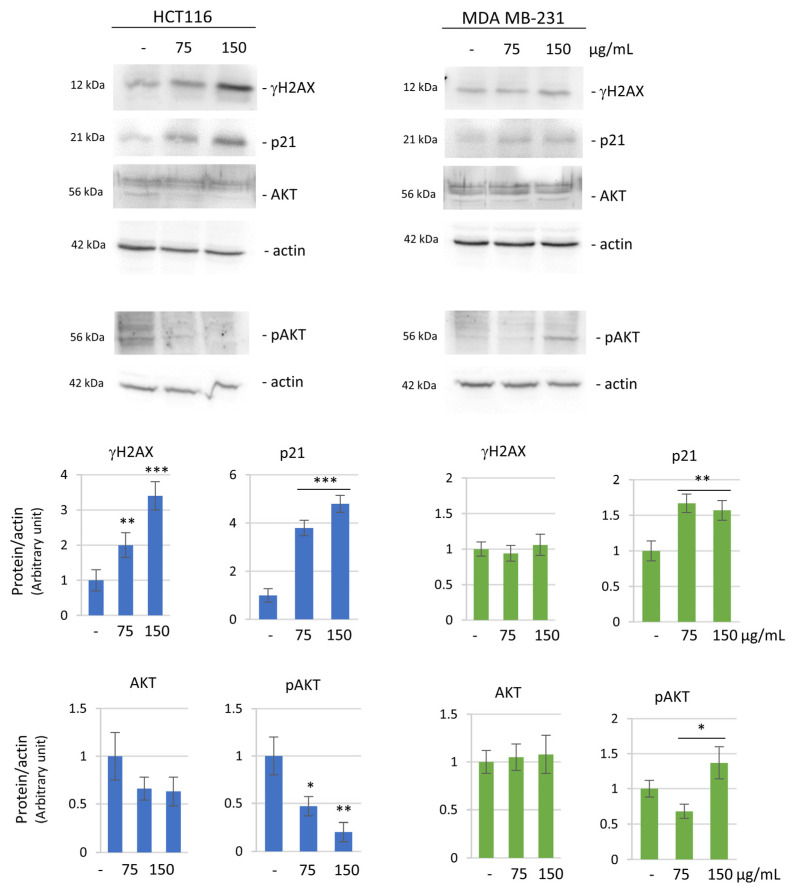
HE effect on the expression of pro-survival markers. Western blotting analysis of γH2AX, p21, AKT and phosphorylated AKT (pAKT) in basal conditions and after 48 h of 75 and 150 µg GAE/mL HE in HCT116 cell (**left** panel) and MDA MB-231 cell (**right** panel). Actin blot was used as a loading control to normalize the protein levels. Densitometric analysis, conducted by Quantity One software, are reported in the histograms below. Results are representative of two independent experiments with similar results and values are expressed as mean ± SD. * *p* < 0.05, ** *p* < 0.01, *** *p* < 0.001 vs. untreated control.

**Figure 5 biomolecules-14-01111-f005:**
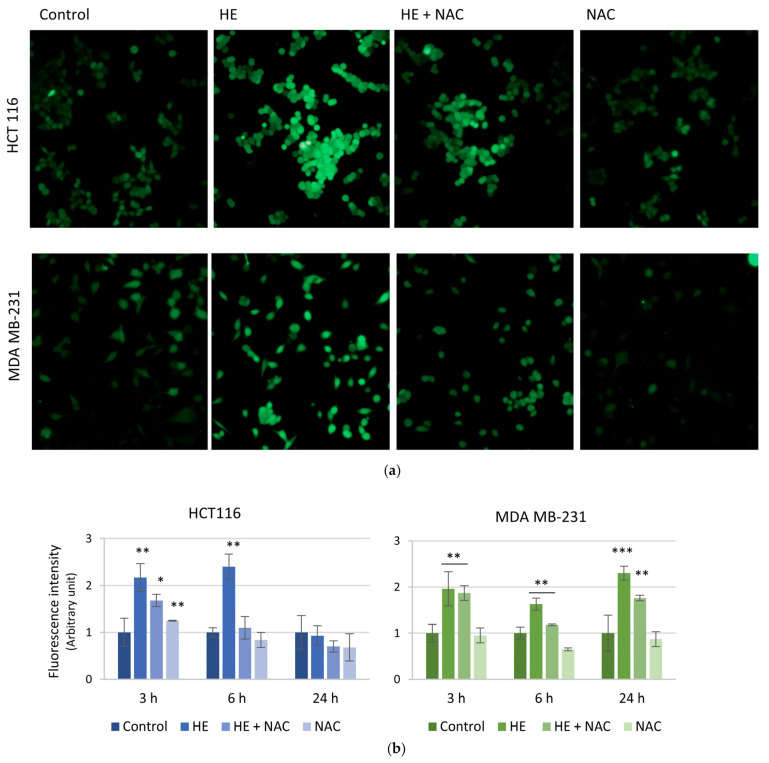
HE effects on reactive oxygen species (ROS) production. HCT116 and MDA MB-231 cells were treated with 150 µg/mL GAE of HE for 3, 6 and 24 h alone or in combination with 5 mM NAC. Magnification 200× (**a**) Fluorescence images are representative of the effects observed after 6 h of treatment. (**b**) Time-dependent analysis of fluorescence intensity was carried out through ImageJ software (version IJ 1.46r). Results are representative of three independent experiments and fluorescence intensity values are expressed as mean ± SD. * *p* < 0.05, ** *p* < 0.01, *** *p* < 0.001 vs. untreated control.

**Figure 6 biomolecules-14-01111-f006:**
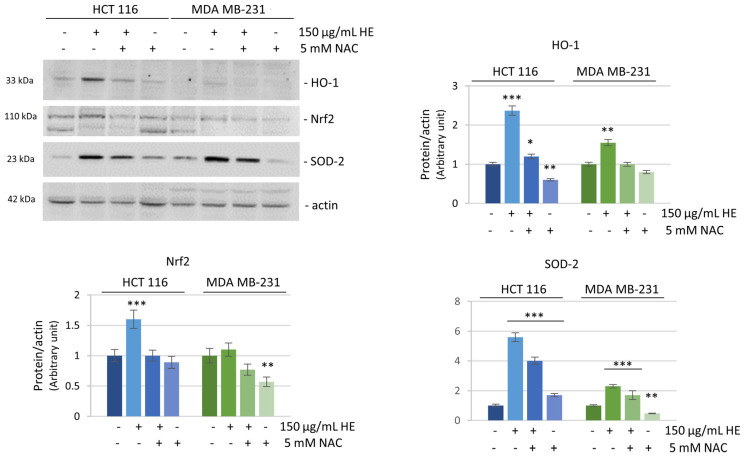
Evaluation of the expression of antioxidant enzymes. Western blotting analysis of some antioxidant proteins (HO-1, Nrf2 and SOD-2) after 150 µg GAE/mL HE treatment for 24 h in HCT116 and MDA MB-231 cells. Protein levels were normalized by actin. Densitometric analysis, conducted by Quantity One software, are reported in the histograms. Results are representative of three independent experiments and values are expressed as mean ± SD. * *p* < 0.05, ** *p* < 0.01, *** *p* < 0.001 vs. untreated control.

**Figure 7 biomolecules-14-01111-f007:**
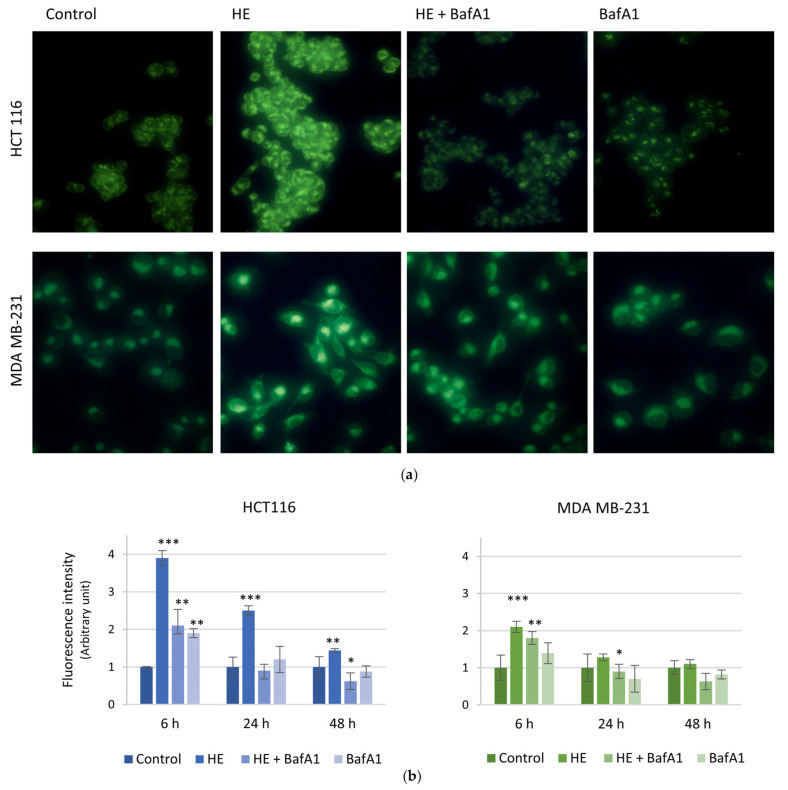
Autophagosome staining after HE treatment in HCT116 and MDA MB-231 cells. HCT116 and MDA MB-231 cells were treated with 150 µg GAE/mL HE alone or in combination with 100 nM bafilomycinA1. After 6, 24 and 48 h the cells were incubated with 50 µM MDC for 10 min and visualized by fluorescence microscope. Magnification 200×. (**a**) Representative images acquired after 6 h of treatment. (**b**) The histogram reports the time-dependent analysis of fluorescence intensity carried out through ImageJ software. Results are representative of two independent experiments with similar results and fluorescence intensity values are expressed as mean ± SD. * *p* < 0.05, ** *p* < 0.01, *** *p* < 0.001 vs. untreated control.

**Figure 8 biomolecules-14-01111-f008:**
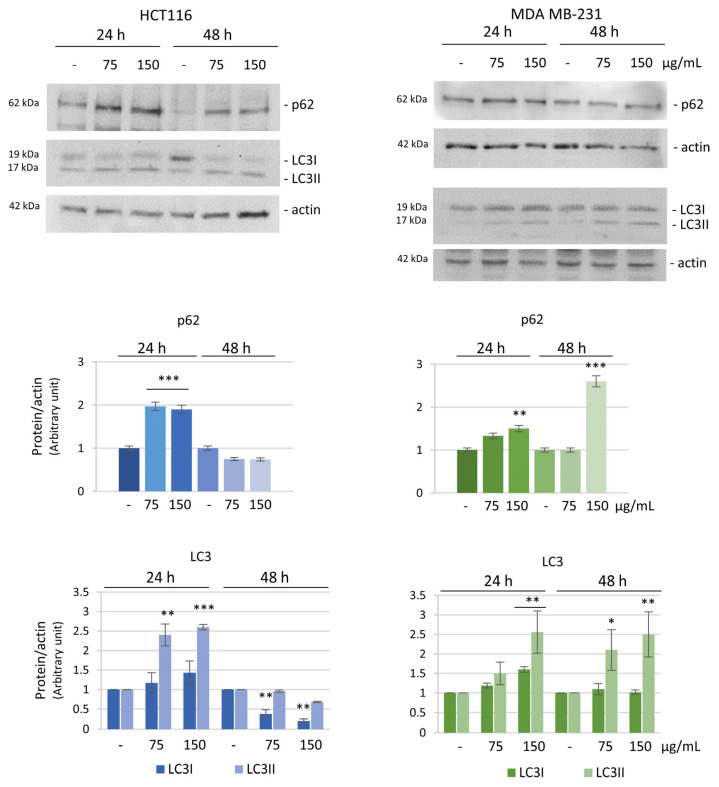
Evaluation of autophagic pathway markers. Western blotting analysis of the autophagic markers p62 and LC3 after 75 or 150 µg GAE/mL HE treatment for 24 or 48 h in HCT116 and MDA MB-231 cells. Protein levels were normalized by actin. Densitometric analysis was conducted by using Quantity One software and are reported in the histograms. Results are representative of three independent experiments and values are expressed as mean ± SD. * *p* < 0.05, ** *p* < 0.01, *** *p* < 0.001 vs. untreated control.

**Figure 9 biomolecules-14-01111-f009:**
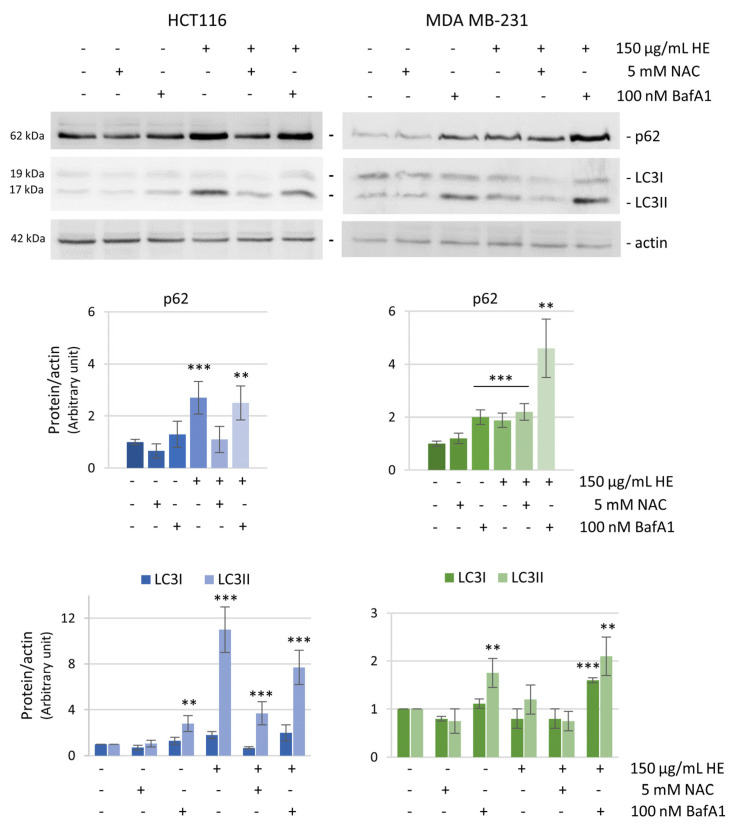
Relationship between ROS and autophagy in HCT116 and MDA MB-231 cells. Western blotting analysis of p62 and LC3 after 24 h of treatment with 5 mM NAC, 100 nM bafilomycin A1 (BafA1) employed alone or in combination with 150 µg GAE/mL HE. Actin blot was used as a loading control to normalize the protein levels. Densitometric analysis, conducted by Quantity One software, are shown by the histograms below. Results are representative of two independent experiments with similar results and values are expressed as mean ± SD. ** *p* < 0.01, *** *p* < 0.001 vs. untreated control.

**Figure 10 biomolecules-14-01111-f010:**
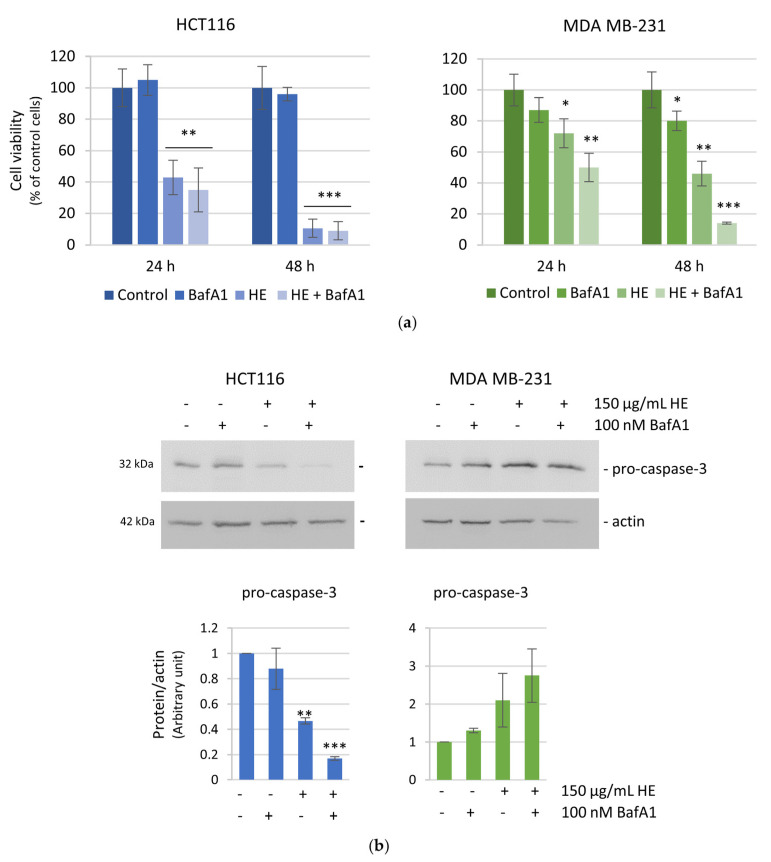
The effect of the autophagy inhibitor bafilomycin A1 (BafA1) on cell viability and apoptotic markers. (**a**) MTT assay was used to assess cell viability after 24 and 48 h of 150 µg GAE/mL HE alone or in combination with 100 nM bafilomycin A1. Results are representative of three independent experiments and are expressed as mean ± SD. * *p* < 0.05, ** *p* < 0.01, *** *p* < 0.001 vs. untreated control. (**b**) Western blotting analysis of pro-caspase-3 after 24 h treatment with 150 µg/mL GAE of HE alone or in combination with 100 nM bafilomycin A1 in HCT116 and MDA MB-231 cells. Protein levels were normalized by actin. Densitometric analysis was conducted by Quantity One software and are reported in the histograms. Results are representative of two independent experiments with similar results and are expressed as mean ± SD. * *p* < 0.05, ** *p* < 0.01, *** *p* < 0.001 vs. untreated control.

**Table 1 biomolecules-14-01111-t001:** Individual compounds identified in the grape pomace extract (HE).

**Phenolic Acid**			
**Analytes**	**R.T.**	**mg/100 g DW**	***m*/*z***
Gallic acid	6.6	20.96	169.01425
Protocatechuic acid	9.0	4.472	153.01824
Caffeic acid	11.4	NF	179.03389
Caftaric acid	9.7	3.22	311.03976
Syringic acid	11.5	NF	197.04445
*p*-coumaric acid	12.8	0.528	163.03897
Ferulic acid	13.0	NF	193.05063
Fertaric acid	11.2	1.728	325.05541
*p*-hydroxybenzoic acid	10.8	1.232	137.02332
**Anthoxanthins (Flavones and Flavonols)**			
**Analytes**	**R.T.**	**mg/100 g DW**	***m*/*z***
Procyanidin B1	8.9	11.488	577.13405
Catechin	9.8	91.96	289.07066
Epicatechin gallate	11.7	5.408	441.08162
Procyanidin B2	9.4	5.792	577.13405
Epicatechin	11.0	52.736	289.07066
Quercetin 3-*O*-glucoside	13.6	5.312	463.0871
Quercetin 3-*O*-glucuronide	14.0	9.08	477.06637
Quercetin 3-*O*-galactoside	12.9	NF	463.0871
Quercetin 3-*O*-rhamnoside	13.3	NF	447.09219
Kaempferol 3-*O*-glucoside	14.4	0.896	447.09219
Quercetin	15.7	20.04	301.03428
Kaempferol	16.8	3.608	285.03936
**Stilbenes**			
**Analytes**	**R.T.**	**mg/100 g DW**	***m*/*z***
trans-Resveratrol	14.9	NF	227.07027
trans-Polydatin	12.3	0.584	389.12309

Retention times (R.T.) are also indicated. Concentrations are expressed in mg/100 g dry weight (DW) and exact masses (*m*/*z*).

## Data Availability

The data that support this study are available from corresponding author upon reasonable request.

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
