# Peer review of "The Antitumor Potential of Sicilian Grape Pomace Extract: A Balance between ROS-Mediated Autophagy and Apoptosis"

_biomolecules, 2024, doi:10.3390/biom14091111_

Round 1

Reviewer 1 Report

Comments and Suggestions for Authors

This study explores the potential antitumor properties of a hydroalcoholic extract (HE) derived from Sicilian grape pomace, focusing on its effects on colon and breast cancer cells. The extract, rich in anthoxanthins and phenolic acids, showed a greater ability to reduce the viability of colon cancer cells compared to breast cancer cells. Although this work is very interesting and the perspective of circular economy is highly commendable, there are certain aspects of the authophagy experiments, which need major revisions.

Autophagy: It was pointed out in the "Cell culture and treatment conditions" that cells were pre-treated with BafA1 one hour before adding HE. Bafilomycin A1 is an inhibitor of the vacuolar-type H+-ATPase, which prevents acidification of lysosomes. Without acidification, lysosomes cannot degrade autophagic cargo. Adding Bafilomycin before the experiment would inhibit the autophagic flux from the beginning, leading to an accumulation of autophagosomes but preventing their degradation. Since autophagosomes accumulate without being degraded, it might falsely appear as if autophagy is upregulated. Furthermore, even when Baf is added, cells will after a while recover and continue with normal autophagy flux. Therefore, pre-treatment would not reflect true autophagic activity but rather a blockade in the autophagy process. Autophagic flux, the process of autophagosome formation and degradation, is a critical measure in autophagy studies. Adding Bafilomycin before the experiment would make it impossible to measure this flux accurately, as degradation is inhibited from the start.  Prolonged inhibition of lysosomal activity can induce cellular stress and potentially trigger apoptosis or necrosis, complicating the interpretation of experimental results. Therefore, Bafilomycin should generally be added toward the end of an autophagy experiment (during the final 1-2 hours of the treatment period) to inhibit lysosomal degradation and allow for the measurement of autophagic flux by comparing autophagosome levels before and after its addition. Adding it too early would lead to misleading results and obscure the true autophagic activity.

It is suggested that these experiments should be repeated while adding Baf at the final 2 hours of the treatment period to make conclusions about autophagy.

Western Blots: Some of the western blots are also of a poor quality which makes it difficult to interpret. 

It is suggested that PKB, both phospho and total is optimized further, by using different Ab concentrations, increasing blocking time, increasing washing steps and exposure time. Please indicate which phosphorylation site has been measured with P-PKB. For Akt, to be fully active, both the Serine (Ser) and Threonine (Thr) residues need to be phosphorylated. Specifically, the phosphorylation occurs at two key residues: Thr308 (Threonine at position 308) in the activation loop of the kinase domain and Ser473 (Serine at position 473) in the hydrophobic motif of the C-terminal tail. Thr308 is typically phosphorylated first. This phosphorylation is carried out by Phosphoinositide-dependent kinase-1 (PDK1) and is essential for partial activation of PKB/Akt. Ser473 is then phosphorylated by mTORC2 (mechanistic target of rapamycin complex 2). This second phosphorylation is required for full activation of PKB/Akt. The phosphorylation of both residues is necessary for PKB/Akt to achieve its full kinase activity and carry out its downstream signaling functions effectively.

The authors should also consider trying to detect cleaved caspase-3, it is not always the case that if there is less pro-/ full length caspase 3, that it will translate into more cleaved product at that specific time point. 

Comments on the Quality of English Language

Overall, the quality of the language is acceptable - authors should avoid using jargon such as "nice", "nicely" and "full-blown".

Author Response

Reviewer 1

This study explores the potential antitumor properties of a hydroalcoholic extract (HE) derived from Sicilian grape pomace, focusing on its effects on colon and breast cancer cells. The extract, rich in anthoxanthins and phenolic acids, showed a greater ability to reduce the viability of colon cancer cells compared to breast cancer cells. Although this work is very interesting and the perspective of circular economy is highly commendable, there are certain aspects of the authophagy experiments, which need major revisions.

We thank the Reviewer for appreciating the rationale of our manuscript and for considering it very interesting in the perspective of circular economy. We also appreciate the suggestions that aim to improve the paper quality, and we would like to discuss them point by point as follows.

Point 1: Autophagy

Reviewer: It was pointed out in the "Cell culture and treatment conditions" that cells were pre-treated with BafA1 one hour before adding HE. Bafilomycin A1 is an inhibitor of the vacuolar-type H+-ATPase, which prevents acidification of lysosomes. Without acidification, lysosomes cannot degrade autophagy cargo. Adding Bafilomycin before the experiment would inhibit the autophagic flux from the beginning, leading to an accumulation of autophagosomes but preventing their degradation. Since autophagosomes accumulate without being degraded, it might falsely appear as if autophagy is upregulated. Furthermore, even when Baf is added, cells will after a while recover and continue with normal autophagy flux. Therefore, pre-treatment would not reflect true autophagic activity but rather a blockade in the autophagy process. Autophagic flux, the process of autophagosome formation and degradation, is a critical measure in autophagy studies. Adding Bafilomycin before the experiment would make it impossible to measure this flux accurately, as degradation is inhibited from the start. Prolonged inhibition of lysosomal activity can induce cellular stress and potentially trigger apoptosis or necrosis, complicating the interpretation of experimental results. Therefore, Bafilomycin should generally be added toward the end of an autophagy experiment (during the final 1-2 hours of the treatment period) to inhibit lysosomal degradation and allow for the measurement of autophagic flux by comparing autophagosome levels before and after its addition. Adding it too early would lead to misleading results and obscure the true autophagic activity. It is suggested that these experiments should be repeated while adding Baf at the final 2 hours of the treatment period to make conclusions about autophagy.

Response: We are aware that Bafilomycin blocks the autophagic cargo degradation; our aim was just to study if HE was able to induce the autophagic process and to evaluate the role of autophagy through its inhibition with bafilomycin. Data reported in literature indicate various conditions of treatments with bafilomycin, including co- or pre-treatment as well as the addition in the last hours of treatment. Based on our experience, we set up the conditions of Bafilomycin A1 treatment in tumor cells considering pre-treatment and co-treatment. Our previous data (Celesia et al. Biomedicines 10(8): 1994, 2022; Emanuele S et al. Nutrients 12;10(10):1490) 2018) indicated that pre-treatment with Bafilomycin A1 represents the best condition to assure autophagy inhibition in our systems.

The aim of our study was not the evaluation of the various steps of autophagic flux, but the specific role of autophagy (pro- or anti-survival) in our cell contest. Results reported in Figure 10 indicated that Bafilomycin A1 alone did not induce signs of cell stress/toxicity, while the inhibition of autophagy in HE-treated cells is able to induce cell death in MDA MB-231 cells, underlying the pro-survival role of autophagy. Moreover, as shown in Fig. 9, pre-treatment with Bafilomycin A1 produced p62 accumulation, an index that autophagy is not completed (Emanuele S. et al. Int J Mol Sci 16;21(14):5029, 2020), as well as accumulation of LC3II. Therefore, we consider our results reliable. In our opinion, adding Bafilomycin A1 at the 2 final hours could not guarantee autophagy inhibition. However, to address the comment of the Reviewer, we decided to better define this point in the discussion, (page 19). 

Point 2: Western blots

Reviewer: It is suggested that PKB, both phospho and total is optimized further, by using different Ab concentrations, increasing blocking time, increasing washing steps and exposure time. Please indicate which phosphorylation site has been measured with P-PKB. For Akt, to be fully active, both the Serine (Ser) and Threonine (Thr) residues need to be phosphorylated. Specifically, the phosphorylation occurs at two key residues: Thr308 (Threonine at position 308) in the activation loop of the kinase domain and Ser473 (Serine at position 473) in the hydrophobic motif of the C-terminal tail. Thr308 is typically phosphorylated first. This phosphorylation is carried out by Phosphoinositide-dependent kinase-1 (PDK1) and is essential for partial activation of PKB/Akt. Ser473 is then phosphorylated by mTORC2 (mechanistic target of rapamycin complex 2). This second phosphorylation is required for full activation of PKB/Akt. The phosphorylation of both residues is necessary for PKB/Akt to achieve its full kinase activity and carry out its downstream signaling functions effectively.

Response: We are aware that the quality of AKT western blotting is not optimal; however the result of the variation of the phosphorylated active form is, in our opinion, unambiguous. We thank the Reviewer for the detailed description of PKB/Akt phosphorylation. The phospho-Akt antibody used specifically recognizes Ser473 (Serine at position 473). We provided to include this information in the Materials and Methods section (page 5). As the Reviewer well reported, Thr308 is first phosphorylated by PDK1, whereas Ser473 is then phosphorylated by mTORC2. Therefore, we assume that Ser473 phosphorylation represents an index of PKB/Akt activation. As the Reviewer stated this second phosphorylation is required for full activation of PKB/Akt. We provided to add a detailed description of PKB/Akt phosphorylation/activation in the discussion (page 18).

Reviewer: The authors should also consider trying to detect cleaved caspase-3, it is not always the case that if there is less pro-/ full length caspase 3, that it will translate into more cleaved product at that specific time point.

Response:

We thank the Reviewer for this observation. Please note that the decrease in pro-caspase 3 in HCT116 cells upon HE treatment is dramatic and it is accompanied by PARP1 cleavage (89 kDa), which is a product of caspase 3 activity. However, to address this issue and considering that the cleaved fragment of caspase 3 was slightly detectable in the entire filter, we reloaded the samples and we cut the filter in order to get a better response. In attached file, the Reviewer can see the result of the experiment done in the short time available. The arrows indicate the active caspase-3 fragment at 17 kDa. A sentence about the presence of the active fragment has been inserted into the text (page 9).

Reviewer: Overall, the quality of the language is acceptable – authors should avoid using jargon such as "nice", "nicely" and "full-blown".

Response: Thanks for the suggestion, done.

Reviewer 2 Report

Comments and Suggestions for Authors

Authors:

When reading the submitted article of Sonia Emanuele et al. on the antitumor potential of grape pomace extract, I have found that the theoretical introduction is giving adequate information on the current status of investigation in the field. Experiments were conducted correctly, and the results are presented clearly, giving sufficient information on the results achieved. Discussion and conclusions summarize well the investigation done.

However, I have found certain number of items to be corrected:

 Lines 123-125: The name of the colorimetric assay should be corrected.

Line 162 and further in the text: Heteroatoms in the chemical names should be written in italics. That error appears as numerous one all over the manuscript.

Figure 2: Which units should be read in the table accompanying the Figure 2? The table should be completed.

Line 425: Grammatical error should be corrected.

I recommend to the editor to accept this manuscript after minor revision.

Comments on the Quality of English Language

Minor grammatical errors were observed in the manuscript. However, they should be corrected during the revision of the manuscript.

Author Response

Reviewer 2

When reading the submitted article of Sonia Emanuele et al. on the antitumor potential of grape pomace extract, I have found that the theoretical introduction is giving adequate information on the current status of investigation in the field. Experiments were conducted correctly, and the results are presented clearly, giving sufficient information on the results achieved. Discussion and conclusions summarize well the investigation done.

Response: We warmly thank the Reviewer for the positive evaluation of our manuscript. We are glad it was appreciated and considered suitable for publication.

Reviewer: Lines 123-125: The name of the colorimetric assay should be corrected.

Response: Many thanks, done.

 Reviewer: Line 162 and further in the text: Heteroatoms in the chemical names should be written in italics. That error appears as numerous one all over the manuscript.

Response: Many thanks, done.

Reviewer: Figure 2: Which units should be read in the table accompanying the Figure 2? The table should be completed.

Response: The numbers in the table indicate the percentage of the cells in the different phases of the cell cycle. The table has been completed.  

Reviewer: Line 425: Grammatical error should be corrected.

Response: Many thanks, done.

Round 2

Reviewer 1 Report

Comments and Suggestions for Authors

Thank you for addressing my comments sufficiently.